# Protective Effects of Ferulic Acid on Metabolic Syndrome: A Comprehensive Review

**DOI:** 10.3390/molecules28010281

**Published:** 2022-12-29

**Authors:** Lei Ye, Pan Hu, Li-Ping Feng, Li-Lu Huang, Yi Wang, Xin Yan, Jing Xiong, Hou-Lin Xia

**Affiliations:** 1School of Pharmacy, Chengdu University of Traditional Chinese Medicine, Chengdu 611137, China; 2Chengdu Institute of Chinese Herbal Medicine, Chengdu 610016, China

**Keywords:** ferulic acid, metabolic syndrome, diabetes, hyperlipidemia, hypertension, obesity

## Abstract

Metabolic syndrome (MetS) is a complex disease in which protein, fat, carbohydrates and other substances are metabolized in a disorderly way. Ferulic acid (FA) is a phenolic acid found in many vegetables, fruits, cereals and Chinese herbs that has a strong effect on ameliorating MetS. However, no review has summarized the mechanisms of FA in treating MetS. This review collected articles related to the effects of FA on ameliorating the common symptoms of MetS, such as diabetes, hyperlipidemia, hypertension and obesity, from different sources involving Web of Science, PubMed and Google Scholar, etc. This review summarizes the potential mechanisms of FA in improving various metabolic disorders according to the collected articles. FA ameliorates diabetes via the inhibition of the expressions of PEPCK, G6Pase and GP, the upregulation of the expressions of GK and GS, and the activation of the PI3K/Akt/GLUT4 signaling pathway. The decrease of blood pressure is related to the endothelial function of the aortas and RAAS. The improvement of the lipid spectrum is mediated via the suppression of the HMG-Co A reductase, by promoting the ACSL1 expression and by the regulation of the factors associated with lipid metabolism. Furthermore, FA inhibits obesity by upregulating the MEK/ERK pathway, the MAPK pathway and the AMPK signaling pathway and by inhibiting SREBP-1 expression. This review can be helpful for the development of FA as an appreciable agent for MetS treatment.

## 1. Introduction

Metabolic syndrome (MetS), also called Reaven syndrome, CHAOS, syndrome X, the deadly quartet and insulin resistance syndrome, is a common metabolic disorder [1]. It is defined by a cluster of severe risk factors, including diabetes, hyperlipidemia, hypertension and obesity [2]. These diseases are some of the significant causes of death worldwide. One study pointed out that MetS can significantly increase the risk of developing cardiovascular disease and type 2 diabetes (T2D) [3]. The paramount causes of MetS are human physical inactivity and unhealthy dietary habits (a high-calorie diet) [4]. Nowadays, MetS has become a global health problem. It is estimated that 20–30% of adults worldwide have MetS [5]. Hence, it is imperative to find a suitable strategy to overcome the worldwide problem. Ample evidence has proved that consuming wholegrain foods rich in phenolic acid and dietary fiber have beneficial effects in reducing the risk of MetS and the associated disorders [6].

Before the late 1990s, researchers studying nutrition focused on the oxidation resistance of carotenoids, vitamins and minerals. In recent years, the antioxidant capacities of phenolic acids and their roles in preventing and treating some cancers, diabetes, cardiovascular diseases and inflammation have been supported by research [7]. Phenolic acids, a type of precious natural resource found in vegetables, fruits and cereals, have pronounced effects on improving MetS. They are usually metabolized and circulated in the body as glucuronized, sulfated and methylated metabolites, showing their biological activity [8]. According to recent studies, phenolic acids such as ferulic, caffeic and protocatechuic acid have apparent protective effects on high-fructose-diet-induced MetS [9,10]. Phenolic acids play a significant role in the therapy for MetS. These compounds ameliorate MetS by different mechanisms, such as improving pancreatic *β*-cell functionality, stimulating insulin secretion, enhancing glucose uptake, and so on [11].

Among the phenolic acids, ferulic acid (FA, 4-hydroxy-3-methoxy cinnamic acid, Figure 1) is a crucial phytochemical found in a wide array of foods such as vegetables, fruits and cereals [12,13]. Furthermore, FA is also found in many Chinese herbal medicines such as Chuanxiong Rhizoma, Angelicae Sinensis Radix and Radix Astragali [14,15,16]. Studies have shown that FA has abundant pharmacological activities and presents an array of therapeutic effects in the treatments of various diseases such as cancer [17], cardiovascular diseases [18], inflammation [19], diabetes [20], neurological disorders [21], liver injuries [22], lung injuries [23], tumors [24], etc. In addition, FA has beneficial effects on improving insulin sensitivity and lipid profiles, reducing blood pressure, preventing the vascular remodeling of mesenteric arteries, and enhancing vascular function [25]. This paper reviews the mechanisms of FA for the improvement of the common symptoms of MetS, and their specific mechanisms are shown in Table 1.

## 2. Literature Search

This review collected articles from different source engines including Web of Science, PubMed, Google Scholar, etc. The following keywords were used in the search: “metabolic syndrome”, “phenolic acid”, “ferulic acid”, “diabetes”, “hyperglycemia”, “blood glucose”, “hypertension”, “blood pressure”, “endothelial function”, “hyperlipidemia”, “serum lipids”, “lipid spectrum”, “dyslipidemia”, “obesity” and “weight.”

## 3. Ferulic Acid and Diabetes

Diabetes, a severe risk factor of MetS, is a chronic metabolic disease that affects a large number of people around the world. It was reported that about 700 million adults worldwide may have diabetes in 2045 [26]. Diabetes usually causes various complications such as diabetic cardiomyopathy, cerebellum lesion, spleen damage, etc. [27,28,29]. The development of diabetic complications is a multifactorial process, and oxidative stress may play a central role [30]. Diabetes can cause the production of free radicals, which can further lead to oxidative stress [31]. Interestingly, some phenolic acids, such as FA, that are present in Chinese herbs play a crucial role in treating diabetes [32]. FA can scavenge oxidative free radicals and inhibit the production of the reactive oxygen species (ROS), thereby ameliorating diabetes [33].

### 3.1. The Animal Studies

Diabetes is divided into insulin-dependent diabetes mellitus (IDDM) and non-insulin-dependent diabetes mellitus (NIDDM). An FA treatment can inhibit the blood glucose levels in both STZ-induced diabetic mice (IDDM model) and KK-Ay mice (NIDDM model) [34]. The administration of FA markedly improved the blood glucose levels, the total serum cholesterol, the triglycerides, the creatinine, the urea and the albumin [35]. In addition, the oral administration of FA (25 and 50 mg/kg) significantly reduced the levels of blood glucose and serum leptin, inhibited insulin resistance and increased the level of serum adiponectin in high-fat diet- (HFD) induced obese mice [36]. FA can also significantly improve the glucokinase activity and hepatic glycogen synthesis [37]. Furthermore, FA can protect against oxidative stress and may prevent hepatocyte and myocardial injuries in obese rats with late-stage diabetes. These effects might be related to the inducement of the increase of antioxidant activities in the plasma, liver and heart and the up-regulation of the mRNA expressions of haem oxygenase-1 (HO-1) and glutathione S-transferase (GST) in the livers and heart cells of diabetic animals [38]. The feruloyl derivatives of FA are also valuable for treating diabetes. For example, feruloylated oligosaccharides extracted from maize bran were beneficial for the early treatment of diabetes [39]. Furthermore, the forming of complexes is probably capable of enhancing the bioactivity of FA. One study investigated the antidiabetic and anti-oxidative synergism between zinc (II) and FA through complexation. The zinc (II) complexation significantly enhanced the bioactivity of FA, including the inhibiting of lipid peroxidation, improving glucose uptake activity, etc. Notably, an in vitro cellular trial demonstrated that the complex was not hepatotoxic or myotoxic [40]. The formation of complexes and derivatives may provide valuable references for the future activities of research on FA.

Skeletal muscle and adipose tissues are two important target sites of insulin. The antidiabetic effect of FA was probably mediated by the facilitating muscle glucose uptake, the inhibition of carbohydrate and lipid hydrolyzing enzymes, and the modulation of oxidative-mediated dysregulated metabolisms [41]. Lipid-induced insulin resistance in skeletal muscle and adipose tissues plays a significant role in the development of T2D. Treatment with FA inhibited insulin resistance induced by saturated fatty acid in the skeletal muscle [42], which might be related to enhancing the protein expression of the insulin receptor substrate -1 in the skeletal muscle [43]. The available commercial oral antidiabetic drugs are known to have some severe side effects. To the best of our knowledge, most phenolic acids are beneficial to improving diabetes. Bean et al. believed that the combination of FA, resveratrol and epigallocatechin-3-O-gallate can improve carbohydrate metabolism. Interestingly, the combination enhances muscle and hepatic insulin resistance more effectively than commercially available antidiabetic drugs such as metformin [44]. In addition, FA and hypoglycemic drugs such as metformin act synergistically in diabetic rats. The synergistic interaction is due to their roles in acting on different pathways to reduce blood glucose [45]. An animal experiment showed that metformin, together with FA (10 mg/kg), can reduce the dose of metformin fourfold (from 50 to 12.5 mg/kg) [45]. Importantly, FA did not produce any adverse effects at high doses or in combination. Furthermore, the combination also increased the number of islets [46].

FA can improve many of the complications induced by diabetes. The administration of FA improved diabetic cardiomyopathy in fructose-streptozotocin-induced T2D [47]. FA alone or in combination with insulin was administered in STZ-induced diabetic rats to treat diabetes-induced neuropathy [48]. In addition, FA intake improves memory via decreasing neuroinflammation and oxidative stress and enhancing insulin sensitivity in diabetic rats with Alzheimer’s disease [49]. Chronic hyperglycemia may cause the excessive production of free radicals, leading to oxidative stress and impairing wound healing. The topical and oral treatment of FA markedly promoted wound healing in diabetic rats [50,51]. The dysfunction of the endothelium plays a significant role in the pathogenesis of vascular disease in diabetes. The co-treatment of diabetic rats by FA and astragaloside IV exerted protective effects against vascular endothelial dysfunction in diabetic rats via the nuclear factor kappa-B (NF-κB) pathway [52]. Diabetic nephropathy was the most severe microvascular complication of diabetes. Both oxidative stress and inflammation were crucial factors in the development of diabetic nephropathy. As an anti-oxidative agent, FA exerted therapeutic effects on diabetic nephropathy by reducing oxidative stress and inflammation [53]. The NOD-like receptor family pyrin domain containing 3 (NLRP3) is a potential target of diabetic nephropathy. The therapeutic effects of FA on diabetic nephropathy might be related to the regulation of the NLRP3 inflammasome signaling pathway [54].

### 3.2. The Human Studies

Diabetic retinopathy, the most common microvascular complication in individuals with diabetes, is closely related to the retinal pigment epithelium (RPE) cells. Zhu et al. investigated the protective effect of FA on the ARPE-19 cells (a human RPE cell line). The FA treatment markedly increased the ARPE-19 cells’ viability and inhibited high glucose-induced cell apoptosis [55]. FA can prevent diabetes-induced vascular dysfunction. One study investigated the protective effects of FA on high glucose-exposed human erythrocytes. The results reported that FA (0.1–100 μM) inhibited lipid peroxidation in erythrocytes and prevented high glucose-induced phosphatidyl serine exposure. Furthermore, FA (10–100 μM) significantly inhibited the levels of glycated hemoglobin in erythrocytes [56]. FA is usually considered as one of the bioactive compounds in wholegrains. A randomized dietary intervention trial demonstrated that adding FA-enriched cereal products to the diet can improve oxidative stress in obese individuals. However, the FA-enriched diet cannot improve fasting and postprandial glucose, lipid metabolism or inflammation parameters [57]. This result revealed the difference between animal experiments and human studies. Therefore, we need further human studies to evaluate the beneficial effects of FA on blood glucose in individuals.

### 3.3. Possible Mechanisms

#### 3.3.1. Inhibition of Expressions of Gluconeogenic Enzymes

As gluconeogenic enzymes, phosphoenolpyruvate carboxylase (PEPCK) and glucose-6-phosphatase (G6Pase) play a crucial role in hepatic glucose production [58]. Both the concentrations of hepatic glycogen and the activity of glycogen synthase (GS) and glucokinase (GK) were markedly reduced in diabetic animals. In contrast, the activity of glycogen phosphorylase (GP) and the enzymes of gluconeogenesis (PEPCK and G6Pase) were increased [59]. This result may be the reason for the increased gluconeogenesis and glucose production in the livers of diabetic animals. In one study, the protective and therapeutic effects of FA on glucose metabolism were evaluated in high-fat-fed mice. FA significantly suppressed blood glucose levels and G6pase and PEPCK activities, whereas it increased glycogen and insulin concentrations and GK activity [60]. This therapeutic effect of FA may be related to FoxO1 (a transcription factor), which can bind directly to the promoters of the gluconeogenic enzyme genes and activate the glucose production process [61]. To be precise, FA suppressed the expressions of the PEPCK and G6Pase proteins by phosphorylating and inactivating FoxO1 to improve blood glucose levels [62] (Figure 2).

#### 3.3.2. Inhibition of GLUT2 Expression

GLUT2 is a bidirectional glucose transporter expressed in the hepatocytes and the pancreatic islet *β*-cells [63]. GLUT2 is of crucial importance for the control of glucose homeostasis. The over-expression of GLUT2 is found in diabetic mice [64]. FA inhibited the expression of GLUT2 in diabetic rats by affecting the interactions between the GLUT2 gene promoter and the transcription factors involving the sterol regulatory element-binding protein-1c (SREBP-1c), the hepatocyte nuclear factors (HNF) 1*α* and HNF 3*β*, and then produced therapeutic effects on diabetes [63] **(**Figure 2).

#### 3.3.3. Improvement of PI3K/Akt/GLUT4 Signaling Pathway

The glucose uptake in adipocytes and myotubes was mainly mediated by the GLUT4 transporter [45]. An early study revealed that the insulin signal cascade mediated by the phosphatidylinositol 3-kinase (PI3K)/phosphorylated-protein kinase B (Akt)/GLUT4 pathway played a crucial role in glucose uptake in the myocardial tissue [65]. FA increased the GLUT4 translocation to the cardiac membrane by activating the PI3K/Akt/GLUT4 signaling pathway, thereby restoring blood glucose levels in the cardiac tissue of diabetic rats [66,67]. Furthermore, several studies point out that FA increased the PI3K and GLUT4 expression levels via the PI3K-dependant pathway, thereby promoting glucose uptake in the L6 myotubes and the 3T3L1 adipocytes [68,69,70] (Figure 2).

#### 3.3.4. Others

Hyperglycemia may activate NF-κB [71]. The immunoreactivity of NF-κB was significantly increased in alloxan-induced diabetic mice. FA probably exerted the antioxidant and antidiabetic effects by inhibiting the proinflammatory factor, NF-κB [72]. In addition, FA also improved postprandial hyperglycemia via the inhibition of the intestinal *α*-glucosidase [73]. FA could inhibit blood glucose levels by promoting insulin release and repressing hepatic glycogenolysis. However, muscle glucose uptake was inhibited at the same time [74]. Insulin deficiency leads to decreased GSK-3*β* phosphorylation levels and hyperactivity in diabetes [75]. The protective effects of FA on the islet *β*-cells and the placental tissues were evaluated in rats with gestational diabetes mellitus (GDM) at the dose of 20 mg/kg body weight for 12 weeks. FA improved GDM by increasing the expressions of p-IRS1, p-IRS2, p-PI3K, GLUT1, GLUT3, GLUT4, etc., promoting the protein expression of visfatin and decreasing apoptosis in the islet *β*-cells [76]. Furthermore, FA probably regulated blood glucose levels by inhibiting the formation of *β*-amyloid fibrils and destabilizing the preformed *β*-amyloid in the pancreas [77].

## 4. Ferulic Acid and Hypertension

Hypertension, a risk factor of MetS, is defined as a systolic reading greater than 140 mmHg or a diastolic reading greater than 90 mmHg. High blood pressure is a risk factor for heart failure, atrial fibrillation, heart valve diseases, chronic kidney disease, aortic syndromes and dementia [78]. There are currently many antihypertensive drugs, including beta-blockers, diuretics, angiotensin II receptor antagonists, angiotensin-converting enzyme inhibitors, and so on [79]. However, lowering the blood pressure excessively with antihypertensive agents increases the incidence of cardiovascular disease [80]. One study demonstrated that FA, an effective component of Chinese herbs, is beneficial in preventing hypertension with no influence on normotension or cardiac function [81]. Furthermore, FA can enhance the antihypertensive effects of antihypertensive drugs such as telmisartan [82].

### 4.1. The Animal Studies

Diabetes is usually accompanied by elevated diastolic and systolic blood pressure, an enhanced contraction response of the aorta to KCl, and a reduced relaxation reaction to acetylcholine (Ach). Furthermore, diabetes is also accompanied by the marked infiltration of leukocytes in the aortic adventitia, endothelial cell pyknosis and the abnormal formation of nitric oxide (NO) and ROS [83]. FA improved the acetylcholine-induced endothelium-dependent vasodilation in spontaneously hypertensive rats (SHR), but not in normotensive Wistar Kyoto (WKY) [84]. In addition, FA also greatly improved plasma NO, glucose tolerance and urinary 8-hydroxy-20-deoxyguanosine in SHR [85]. Pharmacokinetic studies showed that FA is converted predominantly into an FA-4-O-sulfate (FA-sul) in the gut and liver. Interestingly, it may not be that FA, but that FA-sul promoted the vasorelaxation of the saphenous and femoral arteries and aortae [86]. Furthermore, the intravenous injection of FA-sul decreased the mean arterial pressure, whereas FA had no effect [86]. These results may indicate that FA-sul, a metabolite of FA, played an antihypertensive role. One study evaluated the vasoreactivity of FA in chronic two-kidney, one-clip (2K1C) renal hypertensive rats. FA enhanced N-nitro-L-arginine methyl ester (L-NAME)-induced contractile responses and improved the endothelium-dependent vasodilation induced by acetylcholine in 2K1C rats [87]. FA can also enhance endothelial function in isolated aortic rings and H_2_O_2_-treated endothelial cells [88]. The vascular functions, such as vascular reactivity and stiffness, deteriorate with aging. The vasorelaxation of FA is enhanced with aging in SHR, and the vasodilation behavior is independent of the endothelial cells [89]. The therapeutic effects of FA on the cardiovascular complications of diabetes were explained in one study using fructose-induced T2D as an animal model. After 12 weeks of FA treatment, the elevated diastolic blood pressure associated with being fructose-fed was decreased significantly. In addition, FA restored the normal relaxation response of the isolated aortas to Ach [83,90]. It has been confirmed that hypertensive rats are accompanied by left ventricular hypertrophy, increased diastolic stiffness, and heart and kidney fibrosis. FA treatment reduced systolic blood pressure, alleviated left ventricular diastolic stiffness, and reduced ferric iron accumulation, inflammatory cell infiltration and collagen deposition in the left ventricles and kidneys [91].

### 4.2. Possible Mechanisms

#### 4.2.1. Improvement of Endothelial Function

Early studies revealed that NO might be produced from the vascular endothelium and play an essential role in regulating blood pressure and vascular tone [92,93]. It has been proved that endothelial dysfunction is closely related to vascular disorders such as hypertension [94]. Therefore, the impaired endothelium relaxation was possibly caused by the inhibition of the NO vascular activity [95]. FA restored the endothelial function via the enhancement of the bioavailability of NO in SHR aortas [84,86,87]. Furthermore, FA can ameliorate endothelium-dependent relaxation in isolated thoracic aortic rings [85].

#### 4.2.2. Renin-Angiotensin-Aldosterone System

Renin-angiotensin-aldosterone system (RAAS), as a cornerstone in hypertensive therapy, plays a critically important role in the treatment of hypertension [96]. The angiotensin-1-converting enzyme (ACE) is used to catalyze angiotensin I to angiotensin II. It was reported that angiotensin II could increase blood pressure [97]. Therefore, the inhibition of the activity of ACE can result in a decrease in blood pressure. The effects of a single administration of FA on blood pressure were evaluated in one study. The results demonstrated that the ACE activity in the plasma was at the lowest value after 2 h administration [98].

## 5. Ferulic Acid and Hyperlipidemia

Hyperlipidemia, one of the common symptoms of MetS, is characterized by high cholesterol, high triglycerides, or both [99]. It can be either primary or secondary to other diseases [100]. Studies showed that abnormal levels of specific plasma lipids and lipoproteins, such as low-density lipoprotein and cholesterol, could induce cardiovascular diseases [101]. Furthermore, postprandial hyperlipidemia can be atherogenic [102]. At present, statins are the primary means of lipid-lowering therapy. However, statin therapy has many well-recognized side effects [103]. As an active component of many Chinese herbs, FA can significantly reduce the increased plasma triglycerides (TG) and the total cholesterol (TC) and increase the decreased plasma high-density lipoprotein cholesterol (HDL-C) [25].

### 5.1. The Animal Studies

Adding FA to the diet can significantly reduce serum cholesterol in rats [104]. In addition, dietary FA supplements enhanced antioxidant activity in rats and promoted neutral and acidic sterol excretion [105]. This action decreased dietary cholesterol absorption and reduced the plasma and hepatic cholesterol. Interestingly, the hypolipidemic effect of a low dose of FA was more evident than that of a high dose [106]. FA (25 and 50 mg/kg) also reduced the levels of serum lipid, liver cholesterol and TG in obese mice [36]. The combined treatment of ferulic and ascorbic acids restored the elevated the levels of TG, TC, LDL-C and VLDL-C and lowered HDL-C levels in the serum of isoproterenol-intoxicated rats [107]. In addition, the co-treatment of C57 mice with ferulic and caffeic acids improved hyperglycemia, hypercholesterolemia and hypertriglyceridemia. Ferulic and caffeic acid treatments prevented hepatic steatosis by increasing the cholesterol uptake and reducing the hepatic TG synthesis in the liver [108]. FA treatment can reduce increased circulatory lipids (cholesterol, TG, free fatty acids and phospholipids) caused by nicotine [109]. Moreover, FA exhibited a lipid-lowering effect in steatotic FaO rat hepatoma cells, an in vitro model resembling non-alcoholic fatty liver disease (NAFLD) [110]. Excessive TG accumulation in hepatocytes may induce NAFLD, a common liver disorder worldwide. FA can effectively prevent NAFLD by promoting energy expenditure and reducing the accumulation of TG in the liver [111]. The administration of FA inhibited the lipase enzyme activities by eliminating the bile acids and thereby restoring the lipid profile [112]. The hypolipidemic activity of FA was evaluated in HFD-induced hyperlipidemic rats. FA significantly restored the abnormal levels of the hepatic lipid profile, which were HFD-induced. Meanwhile, the administration of FA significantly reduced the elevated levels of oxide-nitrosative stress in the liver [113].

### 5.2. The Human Studies

The incubation of the whole blood and the isolated erythrocytes of hypercholesterolemic patients with FA can partly reduce the cholesterol concentration in erythrocytes [114]. Furthermore, in a randomized, double-blind and placebo-controlled trial, hyperlipidemic patients were randomly divided into two groups (the FA treatment group and the placebo group). FA (1000 mg/d) markedly reduced TC, LDL-C and TG levels and increased HDL-C levels. Moreover, the study also indicated that FA improved lipid profiles, oxidative stress, oxidized LDL-C and inflammation in hyperlipidemic patients, thereby potentially decreasing the risk factors for cardiovascular disease [115]. Adipocytes were cultured either under the “lipid storage state”(the conditions of maintaining but not increasing stored lipids) or under the “ongoing lipogenic state”(the conditions of actively synthesizing and accumulating additional lipids through lipogenesis). In the “lipid storage state”, the co-treatment by low concentrations of FA and quercetin significantly reduced the lipid content, modified the composition of the lipids, and regulated the lipid metabolism genes. Mechanism research showed that these effects were related to PPAR *α*/RXR *α* involvement. In the “ongoing lipogenic state”, FA and quercetin attenuated stored lipid content with a tenfold higher concentration [116].

### 5.3. Possible Mechanisms

The hydroxymethylglutaryl coenzyme A reductase (HMG-Co A reductase), a rate-limiting enzyme in the process of cholesterol synthesis by the hepatocytes, controlled cholesterol synthesis and regulated the expression of lipogenic genes in the liver. FA may reduce TC and LDL-C levels by inhibiting the HMG-Co A reductase. Additionally, FA may decrease lipid peroxidation, resulting in decreased levels of malondialdehyde and oxidized LDL-C [117]. Another study proposed that FA ameliorated lipid metabolism via the decrease of DGAT1 and SCD expressions [67]. Furthermore, the lipid-lowering effects of FA in the liver might be mediated by hepatic lipolysis and fatty acid *β*-oxidation [118]. FA could also ameliorate lipid accumulation partly by inhibiting the expressions of the extracellular signal-regulated kinase (ERK)1/2, c-Jun amino-terminal kinase (JNK)1/2/3 and HGMB1 [119]. Adding FA to the diet can improve fat-induced hyperlipidemia by increasing the fecal lipid excretion and regulating the lipogenic enzyme activities [120]. Long-chain acyl-CoA synthase 1 (ACSL1) is the target of FA in regulating lipid metabolism. FA triggered the mitochondrial membrane distribution of ACSL1, which resulted in the upregulation of adenosine monophosphate (AMP)/adenosine triphosphate (ATP) and, in turn, caused AMPK phosphorylation to inhibit the synthesis of TG and cholesterol [121]. FA improved lipid metabolism by upregulating the AMPK *α* phosphorylation and downregulating the SREBP1 and acetyl-CoA carboxylase 1 (ACC1) expression [122].

## 6. Ferulic Acid and Obesity

Obesity, caused by the accumulation of adipose, is the primary manifestation of MetS. It is characterized by insulin resistance, excessive lipid accumulation, inflammation and oxidative stress [123]. As a global epidemic, obesity is a chronic disorder related to increased morbidity and mortality. At present, the incidence of obesity is rising in many countries [124]. Obesity is usually associated with an increased risk of metabolic dysfunction such as T2D, hypertension and cardiovascular disease [125]. Furthermore, obesity is also related to cardiometabolic disease, malignancy and mental health, etc. [126]. Studies revealed that obesity was related to oxidative stress and chronic low-grade inflammation. The inflammatory sites of obesity are mainly in adipose tissue, the liver, the muscle and the pancreas, etc. [127]. As an active ingredient in herbs, FA has significant antioxidant and anti-inflammatory activities [128]. Therefore, the compound is beneficial to obesity therapy.

### 6.1. The Animal Studies

Exercise is usually considered an effective method of weight control. It was reported that FA was more efficient in reducing lipid deposition in the liver and in the skeletal muscle than exercise, while exercise had a more beneficial effect in improving dyslipidemia. Furthermore, FA combined with exercise showed comprehensive effects in obesity prevention [129]. Therefore, FA combined with exercise might be more suitable for obesity prevention. Oxidative stress plays an important role in the pathophysiology of obesity. Natural compounds such as FA, quercetin and resveratrol improved obesity via their antioxidant activities [130]. The intraperitoneal injection of FA in chicks at a dose of 50 mg/kg reduced their food intake by 70% after 30 min. In addition, FA treatment reduced the defecation frequency of the chicks in a behavior analysis [131]. The phenomenon implied that FA improved obesity by regulating food intake and excretion. Genetic leptin-deficient obese (ob/ob) mice received 0.5% dietary FA for 9 weeks. FA reduced the persistent higher body weights compared to WT mice. Furthermore, after its administration, FA did not affect the total abundance of obesity and anti-obesity-related genes and the diversity of ob/ob mice’s gut microbiota [132]. The gut microbiota is of vital importance for human health. It is reported that an imbalanced gut microbiota composition could induce metabolic diseases such as diabetes and obesity [133]. FA enhances the intestinal barrier by regulating the gut microbiota composition, which may benefit weight control [134]. White adipose tissue is crucial to the inflammatory process related to obesity [135]. The orally administered FA (25 and 50 mg/kg) decreased the lipid droplets in the liver tissues and smaller fat cells in the adipose tissue [36]. Moreover, the administration of FA reduced biomarkers associated with inflammation in the serum of rats [136]. The administration of FA at the concentration of 0.5% in obese C57BL/6J mice reduced daily weight gain, body fat accumulation and white adipose tissue weight, and inhibited the accumulation of hepatic lipids and the release of inflammatory cytokines (IL-6 and TNF-*α*) [137]. The male Swiss HFD mice receiving FA at the dosage of 10 mg/kg for 15 weeks showed significantly reduced adipocyte size, visceral fat accumulation and body weight gain [138]. The anti-obesity potential of many herbs can be attributed to flavonoids such as quercetin glucoside derivatives and phenolics including FA [139]. FA significantly suppressed body weight gain, fat mass increase and dysregulated lipid profiles in the obese male mice [140]. Flours rich in FA and *γ*-oryzanol are beneficial for suppressing weight gain and ameliorating glucose metabolism, hyperlipidemia and lipid accumulation [67]. In addition, the derivatives of FA, such as feruloyl monooleoyl glycerol and feruloyl dioleoyl glycerol, also have anti-obesity potentials [141]. Perhaps the hydroxyl group of FA is essential for weight control.

### 6.2. The In Vitro Studies

One study investigated the potential of FA to regulate adipocyte dysfunction in 3T3-L1 cells by measuring the key adipocyte differentiation markers including glycerol content, lipolysis-associated mRNA and proteins. The results demonstrated that FA significantly inhibited the differentiation of adipocyte and lipid accumulation in 3T3-L1 cells at higher concentrations [123].

### 6.3. Possible Mechanisms

#### 6.3.1. MAPK and MEK/ERK1/2 Signaling Pathways

Mitogen-activated protein kinase (MAPK) consists of ERK, JNK and p38 MAPK. The phosphorylation of p38 MAPK, one of the main pathways of MAPK, can be activated by adipogenic stimulation [82]. The activation of MAPK can inhibit the 3T3-L adipocytic differentiation. Additionally, the deactivation of ERK was essential to inhibiting adipocyte differentiation [142]. The phosphorylation of ERK1/2, activated by the dual specificity kinase MEK1, was revealed to promote the expression of the CCAAT/enhancer-binding protein (C/EBP) *α* and the peroxisome proliferator-activated receptor *γ* (PPAR *γ*) in 3T3-L1 cells [142]. In addition, FA also downregulated key adipogenic transcriptional factors and their downstream targets, such as C/EBP *α*, PPAR *γ* and SREBP1 via upregulating the p38 MAPK and the ERK1/2 signaling pathways, thereby suppressing adipocyte differentiation [143] (Figure 3).

#### 6.3.2. AMPK Signaling Pathways

AMP-activated protein kinase (AMPK) has been proven to be one of the receptors regulating cell metabolism [144]. AMPK plays important roles in adipogenesis [145]. The administration of FA in obese mice prevents weight gain via modulating the AMPK signaling pathways, decreasing adipose inflammation and adipogenesis, and promoting energy dissipation [146]. Furthermore, FA promoted lipolysis by upregulating hormonal-sensitive lipase (HSL) through the beta-adrenergic receptor- (BP) mediated pAMPK *α* activation [143] (Figure 3).

#### 6.3.3. Inhibition of PPAR γ and C/EBP α/β Expression

As adipogenic transcription factors, the PPAR *γ*, C/EBP *α* and C/EBP *β* expressions are essential to adipogenesis [147]. The factors could regulate the expressions of adipocyte-specific genes involving the adipocyte binding protein (aP2), fatty acid synthase (FAS), ACC, lipoprotein lipase (LPL), leptin and adiponectin, which are the key factors of adipogenesis [113]. In an early study, the MAPK-mediated phosphorylation of PPAR *γ* reduced PPAR *γ* transcriptional activity, thereby inhibiting adipocyte differentiation [148]. Furthermore, the FA-treated 3T3-L1 adipocytes exhibited significant anti-adipogenic activity due to the downregulated PPAR *γ* and C/EBP *α* expression [123] (Figure 3).

#### 6.3.4. Others

The self-renewal of the adipose-derived mesenchymal stem cells (ADMSCs) plays a significant role in weight control in obesity. FA contributed to the ADMSCs’ self-renewal and weight control by increasing the population of ADMSCs (Sca-1+CD45-), a hallmark of fat stem cells, and enhancing the NANOG mRNA levels in human ADMSCs [149]. Furthermore, heme oxygenase-1 (HO-1) may disrupt the complex downstream cascade of adipogenesis and lipogenesis of 3T3-L1 cells [150]. The in vitro study evaluated the effect of FA on adipocyte development in the 3T3-L1 pre-adipocyte model. The results demonstrated that FA inhibited adipogenesis by activating HO-1 in 3T3-L1 adipose cells [150,151].

**Table 1 molecules-28-00281-t001:** Mechanisms of ferulic acid improving metabolic syndrome components.

Effects of FA	Experimental Models	Dose/Concentration of FA	Course of Treatment	Results	Mechanisms	References
Anti-hyperglycemia	HFD-induced obese mice	25 and 50 mg/kg	8 weeks	↓ blood glucose level ↓ insulin resistance ↑ the serum adiponectin level	↓ gluconeogenic genes	[36]
	Isolated psoas muscle tissues of rat/α-glucosidase, α-amylase (in vitro)	15, 30, 60, 120 and 240 μg/mL	2 h	↓ blood glucose level	↑ muscle glucose uptake ↓ carbohydrate enzyme activities	[41]
	High fat and fructose-induced T2D rat	50 mg/kg	30 days	↓ blood glucose and serum insulin levels ↑ glucose tolerance and insulin tolerance	↓ gluconeogenesis ↓ negative regulators of insulin signaling ↑ hepatic glycogenesis	[59]
	HFD-induced obese male C57BL/6N mice	High-fat diet supplemented with 0.5% FA	7 weeks	↓ blood glucose level	↓ gluconeogenesis ↑ glucokinase activity ↑ insulin secretion	[60]
	HFD-induced obese C57BL/6 mice	10 mg/kg	12 weeks	↓ blood glucose level	Phosphorylation and inactivation of FoxO1	[62]
	HFD and high fructose water-induced diabetic Wistar rats (in vitro)	50 mg/kg	30 days	↓ hepatic GLUT2 expression	Impairing the interaction between these transcription factors (SREBP1c, HNF1α and HNF3β) and GLUT2 gene promoter.	[63]
	STZ-induced diabetic Wistar rats (in vitro and vivo)	50 mg/kg	8 weeks	↓ blood glucose level ↑ plasma insulin level	↑ phosphorylation of PI3K, Akt, AMPK	[66]
	Differentiated L6 myotubes (in vitro)	25 μM	3 h	↑ uptake of 2-deoxyglucose	Regulation of P13K-dependent pathway	[68]
	3T3-L1 adipocytes (in vitro)	25 μM	24 h	↑ uptake of 2-deoxyglucose	↑ PI3K expression	[69]
	Alloxan-induced diabetic mice	10 mg/kg	15 days	↓ basic biochemical marker (glucose, urea and uric acid, etc.)	↓ the proinflammatory factor, NF-κB	[72]
	HFD-gestational diabetic rats	20 mg/kg	12 weeks	↓ β-cells apoptosis Improvement of insulin signaling	↑ the expression of p-IRS1, p-IRS2, p-PI3K, GLUT1, GLUT3 and GLUT4 ↑ protein expression of visfatin	[76]
	Human amylin peptide (in vitro)	10 μM and 40 μM	6 h	↓ β-cells apoptosis ↑ β-cells mass	↓ islet amyloid cytotoxicity to β-cells	[77]
Anti-hypertension and anti-hyperlipidemia	Thoracic aortic rings from male WKY rats and SHR (in vitro)	10^−5^ to 10^−3^ mol/L	30 min	↑ endothelial function	↑ bioavailability of basal and stimulated NO	[84]
	2K1C hypertensive rats	10^−5^ to 10^−3^ mol/L	30 min	↑ endothelial function	↑ bioavailability of NO	[87]
	Stroke-prone spontaneously hypertensive rats	9.5 mg/kg	6 h	↓ blood pressure	↓ ACE activity in the plasma	[98]
	Diet-induced hypercholesterolemia rats	high-cholesterol diet supplemented with 0.013% FA	5 weeks	↓ the plasma TG and TC concentrations	↓ HMG-Co A reductase	[117]
	Diet-induced hypercholesterolemia weaned piglets	diet supplemented with 0.05% and 0.45% FA	5 weeks	↑ lipid metabolism	↑ lipolysis and fatty acid oxidation	[118]
	Oleic-acid-treated HepG2 cells (in vitro)	0, 12.5, 25 and 50 μg/ml	24 h	↓ cellular lipid accumulation	↓ ERK1/2, JNK1/2/3 and HGMB1 expression	[119]
	Diet-induced hypercholesterolemia mice	0.5%FA diet	7 weeks	↓ plasma and hepatic TC and TG concentrations ↓ lipid peroxidation rate ↓ high-density lipoprotein cholesterol level	↑ fecal lipid excretion Regulation of lipogenic enzymes activities	[120]
	Eight-week-old male db/db diabetic mice	25, 50 and 100 mg/kg	7 days	↑ lipid metabolism	Trigger of the mitochondrial membrane distribution of ACSL 1	[121]
	HFD-induced ApoE^−/−^ mice	40 mg/kg	12 weeks	↑ lipid metabolism	↑ AMPK α phosphorylation ↓ SREBP 1 and ACC 1 expression	[122]
Anti-obesity	3T3-L1 adipocytes (in vitro)	10 μM	24 h	↑ release of glycerol content ↓ lipogenic activities	↓ PPAR γ, C/EBP α and FAS expression ↑ lipolysis-related factors	[123]
	3T3-L1 adipocytes (in vitro)/HFD-induced obese mice	0.2–2 mM/25 and 50 mg/kg	10 days/90 days	↓ cellular lipid accumulation ↓ adipogenesis and lipid accumulation ↓ body weight gain	↓ key transcriptional factors expression ↑ p38MAPK and ERK1/2 signaling pathways Activation of pAMP-α to upregulate HSL	[143]
	Embryo stem cells (ESCs) and adipose-derived mesenchymal stem cells (ADMSCs) (in vitro)	Diet with ferulic acid (5 g/kg diet)	8 weeks	↑ body weight loss ↑ glucose homeostasis, lipid profiling and hepatic steatosis	↑ ADMSCs self-renewal	[149]
	3T3-L1 adipocytes (in vitro)	25, 50 and 100 μM	8 days	↓ adipogenesis	↑ HO-1 expression	[150]

Note: ↑: stimulate, increase, enhance; ↓: inhibit, decrease, reduce; NI: no information; HFD: high fat diet; T2D: type 2 diabetes; STZ: streptozotocin; PI3K: phosphatidylinositol 3-kinase; Akt: phosphorylated-protein kinase B; AMPK: AMP-activated protein kinase; NF-κB: nuclear factor kappa-B; WKY: Wistar Kyoto; SHR: spontaneously hypertensive rats; NO: nitric oxide; 2K1C: two-kidney, one-clip; ACE: Angiotensin-1-converting enzyme; HMG-Co A: Hydroxymethylglutaryl coenzyme A reductase; ERK: extracellular signal-regulated kinase; JNK: c-Jun amino-terminal kinase; ACSL1: long-chain acyl-CoA synthase 1; SREBP1: sterol regulatory element-binding protein 1; ACC 1: acetyl-CoA carboxylase 1; PPAR γ: peroxisome proliferator-activated receptor *γ;* C/EBP α: CCAAT/enhancer-binding protein α; FAS: fatty acid synthase; MAPK: Mitogen activated protein kinase; HSL: hormonal sensitive lipase.

## 7. Conclusions and Future Perspectives

As an active component, FA is found in most plants and its pharmacological effects are extremely extensive. This review summarized the latest research on FA in treating MetS, including symptoms such as diabetes, hyperlipidemia, hypertension and obesity. Furthermore, this paper summarized the possible mechanism of FA in alleviating the above symptoms. FA decreased blood glucose levels mainly via the downregulation of FoxO1 to suppress the gluconeogenic enzyme activities, the activation of the P13K/Akt/GLUT4 signaling pathway to increase glucose uptake, the enhancement of glycogenesis and the inhibition of glucose output in the liver. Moreover, FA probably improves diabetes by inhibiting the proinflammatory factors and intestinal *α*-glucosidase and by reducing the apoptosis in islet *β*-cells. The decrease in blood pressure is related to the endothelial function of the aorta and RAAS. FA reduces blood pressure via the enhancement of the bioavailability of NO to improve endothelial function and the suppression of ACE activity in the plasma. FA improves the lipid spectrum by suppressing the HMG-Co A reductase, promoting the ACSL1 expression, and regulating the factors associated with lipid metabolism. Furthermore, the anti-obesity effect of FA is mediated via upregulating the MEK/ERK and MAPK signaling pathways to inhibit the expressions of PPAR *γ*, C/EBP *β* and C/EBP *α*, upregulating the AMPK signaling pathways to promote HSL expression, and inhibiting the SREBP-1 expression.

According to pharmacokinetic studies, FA is rapidly absorbed and eliminated with low bioavailability after a single oral administration. This may be a challenge which needs to be addressed in the future. It has been confirmed that the synergistic effect of drugs improves the bioavailability of FA. Therefore, we believe that finding synergistic drugs to enhance the bioavailability of FA will be the focus of future research. In addition, bioavailability is often directly related to the chemical structures of drugs. Therefore, the bioavailability of FA can also be changed through structural modification in future research. FA can effectively improve the common symptoms of MetS in animal studies. However, the therapeutic effects of FA are not satisfactory in human studies. Therefore, enhancing the therapeutic effects of FA on MetS is also a great challenge for future research. The studies presented revealed some methods to improve the therapeutic effects of FA on MetS. The formation of complexes and derivatives seems to be an effective way.

It was reported that exercise and FA treatment were beneficial to improving obesity in a synergistic manner. Therefore, MetS might be ameliorated mainly by changing human lifestyles and partly by medical therapy. Unfortunately, according to the collected articles, it is spotted that there are only a few studies performed on humans, which undoubtedly limits the clinical application of FA. It is indicated that clinical research on FA improving metabolic syndrome is still in the embryonic stage at present. For further clinical application of FA, the clinical guidance of FA on MetS needs to be supported by more human studies in the future.

## Figures and Tables

**Figure 1 molecules-28-00281-f001:**
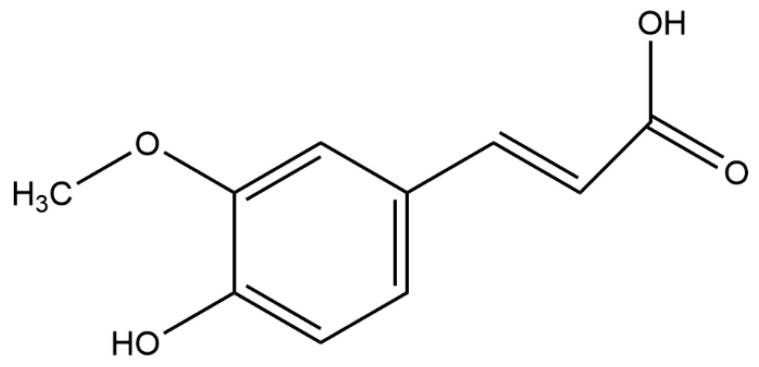
Chemical structure of ferulic acid.

**Figure 2 molecules-28-00281-f002:**
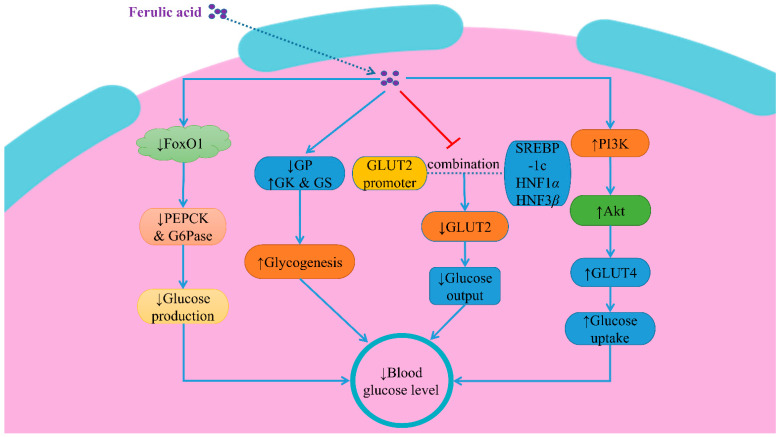
The molecular mechanisms of ferulic acid in improving type 2 diabetes. Note: ↑: activate/upregulate; ↓: inhibit/downregulate; PI3K: phosphatidylinositol 3-kinase; Akt: phosphorylated-protein kinase B; PEPCK: Phosphoenolpyruvate carboxylase; G6Pase: glucose-6-phosphatase; GP: glycogen phosphorylase; GK: glucokinase; GS: glycogen synthase; SREBP1c: sterol regulatory element-binding protein-1c; HNF: hepatocyte nuclear factor.

**Figure 3 molecules-28-00281-f003:**
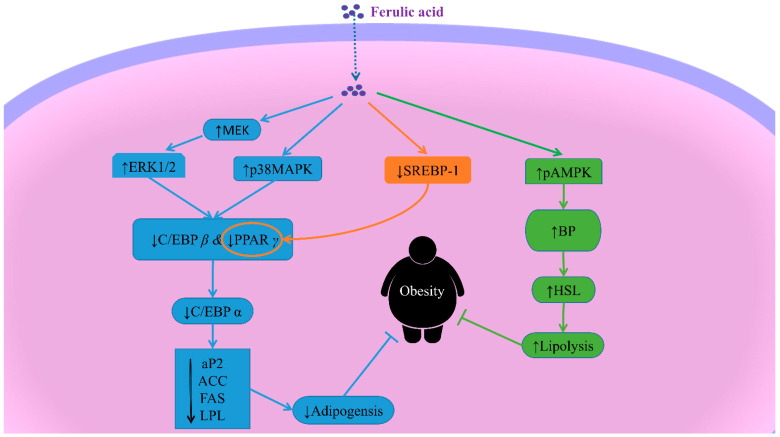
The molecular mechanism of ferulic acid improving obesity. Note: ↑: activate/upregulate; ↓: inhibit/downregulate; ERK: extracellular signal-regulated kinase; MAPK: Mitogen activated protein kinase; PPAR *γ*: peroxisome proliferator-activated receptor *γ*; C/EBP: CCAAT/enhancer-binding protein; SREBP-1: sterol regulatory element-binding protein-1; ACC: acetyl-CoA carboxylase; aP2: adipocyte binding protein; FAS: fatty acid synthase; LPL: lipoprotein lipase; AMPK: AMP-activated protein kinase; BP: beta-adrenergic receptor.

## Data Availability

Not applicable.

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
