# Peer review of "Protective Effects of Ferulic Acid on Metabolic Syndrome: A Comprehensive Review"

_molecules, 2022, doi:10.3390/molecules28010281_

Round 1
Reviewer 1 Report
The manuscript entitled “Protective effects of ferulic acid on metabolic syndrome: a com-prehensive review” by Ye et al. has reviewed the effect of ferulic acid (FA).
Authors sited over 150 papers. However, it is uncomfortable for me that many references were not properly cited.
For example, you mentioned “MetS has become more prevalent worldwide, mainly due to physical inactivity and unhealthy dietary habits [4]” (Page 1). However, I couldn't find the content corresponding to your sentence in ref. 4. Paper focused childhood obesity.
Moreover, your mentioned “It was reported that about 700 million adults may have diabetes in 2045 [25]” (Page 3) referred #25 (Zhang, Y.; Wang, T.; Hu, X.; Chen, G. Vitamin a and diabetes. J. Med. Food 2021, 24, 775-785. http://dio.org/10.1089/jmf.2020.0147). No mention about the number of DM patients in the abstract section, and it is impossible to access http://dio.org/10.1089/jmf.2020.0147. Although I did show two examples, there are many papers sited unproperly.
Also, there are numerous grammatical and spelling errors.
It would be nice to summarize papers related to your research before starting exepriments.
However, in my personal opinion, without adding your own research evidence, you could not reach to write this type of review.
So, unfortunately, your manuscript should be rejected.
Author Response
Q1 Authors sited over 150 papers. However, it is uncomfortable for me that many references were not properly cited.
For example, you mentioned “MetS has become more prevalent worldwide, mainly due to physical inactivity and unhealthy dietary habits [4]” (Page 1). However, I couldn't find the content corresponding to your sentence in ref. 4. Paper focused childhood obesity.
Moreover, your mentioned “It was reported that about 700 million adults may have diabetes in 2045 [25]” (Page 3) referred #25 (Zhang, Y.; Wang, T.; Hu, X.; Chen, G. Vitamin a and diabetes. J. Med. Food 2021, 24, 775-785. http://dio.org/10.1089/jmf.2020.0147). No mention about the number of DM patients in the abstract section, and it is impossible to access http://dio.org/10.1089/jmf.2020.0147. Although I did show two examples, there are many papers sited unproperly.
Response: Thanks for your comments. We are sorry for our mistakes. Because we used Noteexpress to insert references, this software wrongly changed the doi to dio, thus it is unable to access references. We have addressed this issue.
In addition, we have checked the contents of references cited, and we have revised the sentence or reference mainly in Introduction section.
Q2 Also, there are numerous grammatical and spelling errors.
Response: In our revised manuscript, we have invited a colleague who is good at English to address this issue. For example, "at a dose of" was revised to "at the dose of" at many places.
Q3 It would be nice to summarize papers related to your research before starting exepriments.
However, in my personal opinion, without adding your own research evidence, you could not reach to write this type of review.
So, unfortunately, your manuscript should be rejected.
Response: Ferulic acid is one of the major bioactive compounds in Ligusticum wallichii Franch., and the in vivo metabolites of benzoin. Our lab has researched Ligusticum wallichii Franch. and benzoin, including the bioactive compounds identification and screening, and the pharmacology of these compounds including ferulic acid, for years (the reviewer can check it in website: https://www.cnki.net/).
Reviewer 2 Report
The review article has focused on an important topic. The article is quite informative and well organised. I would recommend to accept the paper after addressing the following minor issues.
1. In the introduction section I would recommend authors to incorporate the prevalence of metabolic syndrome (MetS) worldwide.
2. In the introduction, authors need to start the last paragraph with Ferulic acid and add the first 3 lines of the last paragraph at the end of the second paragraph to give emphasis on the ferulic acid and make it more visible and distinct.
3. Novelty of the manuscript is missing. Authors require to justify why they have written this review article.
4. in Table 1, I would strongly suggest authors to include results of the different studies which will prove the potency of ferulic acid in treating/improving MetS.
Author Response
The review article has focused on an important topic. The article is quite informative and well organised. I would recommend to accept the paper after addressing the following minor issues.
Response: We are very pleased to read the reviewer's positive comments on our study while pointing out the drawbacks of our manuscript. We have studied the comments carefully and have revised the manuscript accordingly.
Q1. In the introduction section I would recommend authors to incorporate the prevalence of metabolic syndrome (MetS) worldwide.
Response: Thanks for the suggestion. In the first paragraph of Introduction, we have added the sentences " Nowadays, MetS has become a global health problem. It is estimated that 20%-30% of adults worldwide have MetS [5]. Hence, it is imperative to find a suitable strategy to overcome the worldwide problem."
Q2. In the introduction, authors need to start the last paragraph with Ferulic acid and add the first 3 lines of the last paragraph at the end of the second paragraph to give emphasis on the ferulic acid and make it more visible and distinct.
Response: Thanks for the suggestion. We have done this as suggested.
- Novelty of the manuscript is missing. Authors require to justify why they have written this review article
Response: Thanks for the constructive suggestion. Ferulic acid is a hot compound. Many animal and human-based studies have demonstrated that it is effective in treating metabolic syndrome. However, no review has summarized the mechanisms of FA in metabolic syndrome. Our manuscript is the first review that has addressed this issue. In our revised manuscript, we have added the sentence "However, no review has summarized the mechanisms of FA in treating MetS." in Abstract.
- in Table 1, I would strongly suggest authors to include results of the different studies which will prove the potency of ferulic acid in treating/improving MetS.
Response: Thanks for the constructive suggestion. In our revised manuscript, we have added a column describing the results of the different studies in Table 1.